# Probiotics Function as Immunomodulators in the Intestine in C57Bl/6 Male Mice Exposed to Inhaled Diesel Exhaust Particles on a High-Fat Diet

**DOI:** 10.3390/cells11091445

**Published:** 2022-04-25

**Authors:** Danielle T. Phillippi, Sarah Daniel, Kayla N. Nguyen, Bea Angella Penaredondo, Amie K. Lund

**Affiliations:** Department of Biological Sciences, Advanced Environmental Research Institute, University of North Texas, Denton, TX 76203, USA; daniellephillippi@my.unt.edu (D.T.P.); sarahthomas056@gmail.com (S.D.); kaylanguyen@my.unt.edu (K.N.N.); beaangellapenaredondo@my.unt.edu (B.A.P.)

**Keywords:** intestinal integrity, diesel exhaust particulate matter, mucosal barrier, tight junctions, probiotics, high-fat diet

## Abstract

Epidemiological studies reveal a correlation between air pollution exposure and gastrointestinal (GI) diseases, yet few studies have investigated the role of inhaled particulate matter on intestinal integrity in conjunction with a high-fat (HF) diet. Additionally, there is currently limited information on probiotics in mitigating air-pollutant responses in the intestines. Thus, we investigated the hypothesis that exposure to inhaled diesel exhaust particles (DEP) and a HF diet can alter intestinal integrity and inflammation, which can be attenuated with probiotics. 4–6-w-old male C57Bl/6 mice on a HF diet (45% kcal fat) were randomly assigned to be exposed via oropharyngeal aspiration to 35 µg of DEP suspended in 35 µL of 0.9% sterile saline or sterile saline (CON) only twice a week for 4 w. A subset of mice was treated with 0.3 g/day of Winclove Ecologic^®^ barrier probiotics (PRO) in drinking water throughout the duration of the study. Our results show that DEP exposure ± probiotics resulted in increased goblet cells and mucin (MUC)-2 expression, as determined by AB/PAS staining. Immunofluorescent quantification and/or RT-qPCR showed that DEP exposure increases claudin-3, occludin, zona occludens (ZO)-1, matrix metalloproteinase (MMP)-9, and toll-like receptor (TLR)-4, and decreases tumor necrosis factor (TNF)-α and interleukin (IL)-10 expression compared to CON. DEP exposure + probiotics increases expression of claudin-3, occludin, ZO-1, TNF-α, and IL-10 and decreases MMP-9 and TLR-4 compared to CON + PRO in the small intestine. Collectively, these results show that DEP exposure alters intestinal integrity and inflammation in conjunction with a HF diet. Probiotics proved fundamental in understanding the role of the microbiome in protecting and altering inflammatory responses in the intestines following exposure to inhaled DEP.

## 1. Introduction

Air pollution is a significant environmental health risk and is estimated to account for over 7 million deaths per year globally [1]. Studies have shown that components of air pollution have detrimental effects on multiple organs systems, including the cardiovascular, respiratory, and central nervous systems, among others [2,3,4]. Recent epidemiological studies have shown a clear correlation between air pollution exposure and gastrointestinal (GI) disease characterized by increased hospitalization of patients with inflammatory bowel disease (IBD), colon cancer, and appendicitis. However, the role of air pollution components, such as particulate matter (PM), in mediating these responses in the GI tract has not been fully elucidated [5,6,7,8]. Furthermore, a large portion of the Western world eats a diet high in fat (>30%), contributing to the epidemic of obesity and cardiovascular disease. In addition to these disease states, consuming a high-fat (HF) diet is also associated with increased intestinal permeability, inflammation, and dysbiosis [9,10,11,12]. Currently, there is very little information in the literature characterizing the outcomes of multiple insults, such as air pollution exposure and a HF diet, on gut integrity.

The intestinal epithelial barrier is an important separation between gut microbiota and systemic environment and consists of the mucus layer, epithelial barrier, and submucosal layer that functions to host healthy gut microbiota, transport nutrients, and protect against pathogens and toxins [13]. The mucus layer is often considered the first line of defense against pathogens and toxins. It plays a critical role in maintaining healthy gut microbiota by providing glycans as an energy source for microbes and preventing intestinal microbiota and toxin contact with the epithelial layer [13,14]. Goblet cells are responsible for the synthesis and secretion of mucin, predominately gel-forming MUC2 mucins, which remain bound to the epithelial layer [14]. MUC2^−/−^ mice, which display decreased mucus barrier, have diminished homeostatic function of the intestine and fail to prevent bacterial attachment to the epithelial tissue, associated with colitis and colon cancer [15]. The epithelial layer consists of a single layer of enterocytes supported by a complex connection of tight junction (TJ) proteins, which maintain a cell-to-cell seal and prevent luminal contents from entering the circulation [16,17]. Claudins and occludins are transmembrane TJ proteins that are associated with major intracellular TJ protein, zonula occludens (ZO). Dysregulation of these TJ proteins is associated with increased gut permeability and pathologies, such as IBD, endotoxemia, obesity, and systemic inflammation [16,18].

Matrix metalloproteinases (MMP) are enzymes involved in the breakdown of the extracellular matrix and play a role in tissue degradation and remodeling [19]. MMP-9 is an important mediator of intestinal inflammation and has been shown to increase intestinal epithelial tight junction permeability during activation [20]. MMP-9^−/−^ mice show an increase in goblet cells and MUC2 expression along with increased permeability, suggesting an integral role for MMP-9 in regulating the intestinal epithelial barrier [21,22]. Alterations in TJ proteins and MMP-9 activity can mediate local and systemic pro-inflammatory pathways, such as tumor necrosis factor (TNF)-α activation. Interleukin (IL)-10 is an important anti-inflammatory cytokine in the intestines, and emerging studies have shown its function to maintain intestinal barrier integrity by inhibiting pro-inflammatory cytokine signaling [23]. 

Toll-like receptor (TLR)-4 is a transmembrane protein responsible for the recognition of lipopolysaccharide (LPS), a component of gram-negative bacteria. A HF diet results in increased LPS in the gut and induces TLR-4 signaling pathways [9]. TLR-4 activation by LPS mediates nuclear factor kappa B (NF-κB), which signaling is involved in the regulation of intestinal immune responses [24]. The use of probiotics, which are live microorganisms, has been shown to promote a healthy microbiome by stimulating the immune system, increasing intestinal integrity, providing essential metabolites, and protecting against pathogens [25,26,27,28]. For example, probiotics containing *Bifidobacterium* have been shown to have anti-inflammatory effects by mechanisms such as LPS-mediated NF-κB signaling and can improve intestinal integrity [29,30]. Furthermore, *Lactobacillus* probiotics have been shown to have antipathogenic mechanisms through promoting enhanced epithelial barrier integrity by stimulating mucus production and phosphorylation of TJ proteins [31]. Recent studies have shown that probiotics have local and systemic immunomodulatory effects, regulating TLR expression and immune cell activity in the intestine [32]. Currently, there is a scientific need to understand the function of probiotics in the gut and how they can alter physiological responses during environmental stressors. 

There are three proposed mechanisms by which inhaled PM can alter the gut integrity: (1) by direct effects in the GI tract due to mucociliary clearance of PM from the lungs and subsequent ingestion, (2) by indirect systemic signaling pathways due to inhaled PM being absorbed across the pulmonary membrane, then translocated systemically producing effects via the gut vasculature, and/or (3) by indirect effects where the absorbed PM triggers systemic stressors (i.e., oxidative stress, and inflammatory pathways) that can then promote gut dysbiosis and/or alterations in tissue integrity. While previous studies have characterized the effects of ingested PM on intestinal integrity and gut microbial profiles, few studies have investigated the effects of inhaled PM on the intestinal epithelial barrier. A study by Mutlu et al. found that inhaled ambient PM (135.4 µg/m^3^ for 8 h/d for 3 w) resulted in altered microbiota throughout the GI tract and inflammation in the colon in C57Bl/6 wild-type mice [33]. Our lab investigated the effects of whole-body inhalation exposure to mixed diesel and gasoline engine emission (MVE, 50 µg/m^3^ PM gasoline + 250 µg/m^3^ PM diesel emissions) and woodsmoke (WS, 440 µg/m^3^ PM) for 6 h/d for 50 d and found MVE and WS-exposure resulted in altered gut epithelial integrity, inflammation, and dysbiosis in the small intestine of ApoE^−/−^ mice [34]. 

Importantly, the small intestine consists of three regions: duodenum, jejunum, and ileum, and is responsible for the majority of food digestion and nutrient absorption. Each region of the small intestine has a unique physiology, function, and microbiome, distinctly different from that of the large colon [35]. With most studies identifying fecal microbiome and large intestine integrity, there is a need to characterize changes throughout all regions of the digestive system. To our knowledge, no studies have characterized the effects of inhaled PM or a HF diet ± probiotic treatment on the three regions of the small intestine to date. Thus, we investigated the hypothesis that inhaled diesel exhaust particles (DEP) alter intestinal mucosal and epithelial integrity and promote inflammation, which is further exacerbated by consumption of a HF diet, in C57Bl/6 male mice. Furthermore, we determined whether concurrent oral treatment with a probiotic could mitigate the detrimental effects of exposure and/or a HF diet.

## 2. Methods

### 2.1. Animal Exposures

Some 4–6-w-old C57Bl/6 wild-type male mice (Taconic, New York, NY, USA, C57BL/NTac) on a high-fat (HF) diet consisting of 45% fat (HF, Research Diets #D12451, *n* = 48) were randomly assigned to be exposed via oropharyngeal aspiration (OA) to 35 µg diesel exhaust particles purchased from National Institute of Standards and Technology (NIST, Standard reference material #2975), suspended in 35 µL 0.9% sterile saline (*n* = 24), or sterile saline only (CON, *n* = 24) twice a week for 4 w. The exposure methodology for this study has been previously published by our lab [36,37]. Briefly, animals were housed four to a cage, had access to food and water ad libitum, and were maintained on a 12-h light/dark cycle in humidity and temperature-controlled rooms. The rationale for choosing OA was two-fold: (1) the OA exposure route simulates the route particulate matter would take from the oropharynx region into the lungs, including contact with the mucociliary escalator, which is likely an important source for ingested PM after inhalation exposure and resulting effects on the gut and/or resulting microbiota profiles, and (2) compared to whole chamber inhalation models, OA exposure limits the amount of PM that comes in contact with mouse body and environment, thus reducing the amount that is ingested during grooming and/or eating. As such, for this study, the PM that encounters the digestive system can be assumed to originate from mucociliary clearance and, subsequently, swallowed. All animal procedures were reviewed and approved by the University of North Texas IACUC and to the Guide for the Care and Use of Laboratory Animals (NIH Publication No. 85-23, rev. 1996).

### 2.2. Probiotic Treatment

A subset of mice was dosed with 0.3 g/d of probiotic (~7.5 × 10^8^ CFU/d) Ecologic^®^ Barrier 849 (Winclove Probiotics, B.V., Amsterdam, The Netherlands) in drinking water (PRO, *n* = 12) over the course of the exposure study. Probiotic formulation is a proprietary probiotic blend consisting of nine bacterial strains: *Bifidobacterium bifidum* W23, *B. lactis* W51, *B. lactis* W52, *Lactobacillus acidophilus* W37, *L. brevis* W63, *L. casei* W56, *L. salvarius* W24, *Lactococci lactis* W19, and *Lc. lactis* W58 (‘‘Ecologic^®^ Barrier”, Winclove Probiotics) in a carrier matrix of maize starch, maltodextrin, and minerals. Probiotics were dosed in low-drip metered water bottles and measured daily to determine the average consumption of probiotics per cage. Probiotic administration via drinking water was chosen over oral gavage to minimize stress to mice. Dosage calculation was determined by reference to a previous study using the same probiotic formulation delivered via drinking water in a rodent model, which resulted in significantly decreased inflammatory signaling in rodents fed a high-fat diet [38]. Probiotics were changed daily between 4:00–6:00 p.m.

### 2.3. Tissue Collection

Mice were anesthetized with Euthasol and euthanized by exsanguination via cardiac puncture within 24 h of final exposure. The small intestine was immediately excised and weighed (data not shown). Luminal contents were flushed, collected, snap-frozen, and stored in −80 °C. The small intestine was dissected into three portions: duodenum, jejunum, and ileum. A 2 cm section of each the proximal portion of the duodenum, the proximal portion of the jejunum, and the distal portion of the ileum were collected and immediately fixed in zinc formalin buffer (Sigma–Aldrich, St. Louis, MO, USA, catalog #Z2902). The remaining portions of the small intestine were snap frozen and stored at −80 °C for later analysis. Fixed intestinal portions were dehydrated in ethanol and treated in HistoChoice^®^ Clearing Agent (VWR, Wayne, PA, USA, catalog #97060-932). Tissues were then embedded in paraffin blocks and sectioned at 8 µm.

### 2.4. Histology

For mucin and goblet cell quantification, 8 µm sections of tissue were deparaffinized three times for 5 min using HistoChoice^®^ Clearing Agent (VWR, catalog #97060-932) and rehydrated. Alcian Blue Periodic Acid Schiff (AB/PAS) staining was performed following manufacturer protocol (Fisher Scientific, Waltham, MA, USA, #88043, #88016). Alcian blue stains highly acidic mucopolysaccharides blue and PAS stain neutral-acidic mucopolysaccharides pink. Intestinal goblet cells contain both neutral and acidic mucins which can be determined by a deep purple stain within the intestinal villi structure. Images were blindly scored on a scale from 1–5 for neutral-acidic mucins (pink), highly acidic mucins (blue), and a combination of mucins (deep purple) along the intestinal villi. Scoring was quantified on a 5-point scale based on the intensity of pink, blue, and purple, with 5 being the most intense, and combined for a total of a 15-point scale. Goblet cells were quantified per 100 µm of intestinal villi. A minimum of 4 sections per slide with a minimum of 4–5 locations per section were imaged, and *n* = 3–5 per group were used for analysis. Histological endpoints were imaged at 20x and 40x magnification with a bright field microscope and luminal contents were excluded in the analysis.

### 2.5. Immunofluorescence 

To determine intestinal integrity and inflammation in the intestines, we used immunofluorescent staining to quantify claudin-3, occludin, zonula occludens 1 (ZO-1), tumor necrosis factor α (TNF-α), mucin 2 (MUC2), matrix metalloproteinase 9 (MMP-9), interleukin 10 (IL-10), toll-like receptor 4 (TLR-4), and nuclear factor kappa B (NF-κB). Tissues were stained using the following primary antibodies: claudin-3 (Abcam, Cambridge, MA, USA, #15102; 1:500), occludin (Abcam, #216327; 1:500), ZO-1 (Abcam #59720; 1:250), TNF-α (Abcam, #6671; 1:250), MUC2 (Abcam #11197; 1:250), MMP-9 (Abcam, #38898; 1:1000), IL-10 (Santa Cruz Biotechnology, Dallas, TX, USA, #365858; 1:200), TLR-4 (Santa Cruz Biotechnology, #52962; 1:200), and NF-κB p65 (Abcam, #86299; 1:500). Secondary antibodies used were anti-rabbit Alexa Fluor 555 (Invitrogen, Carlsbad, CA, USA, #A31572) and anti-mouse Alexa Fluor 488 (Invitrogen, #A11101). 8 µm sections of tissue containing all three portions of the small intestine (duodenum, jejunum, and ileum) were deparaffinized three times for 5 min using HistoChoice^®^ Clearing Agent and rehydrated to 70% ethanol. To block autofluorescence, Sudan Black B (SBB; VWR, catalog#100504-304) was used as a quenching agent. SBB was prepared in 70% ethanol to make 0.3% SBB solution and stirred in the dark for 2 h and stored at 4 °C. Tissue was placed in 0.3% SBB for 10 min, then rinsed in 70% ethanol for 1 min. The tissue was then permeabilized in 0.1% Triton X-100 for 5 min and rinsed. Antigen retrieval was performed in boiling EDTA for 5 min and allowed to cool on the benchtop for 5 min. The tissue was then rinsed and blocked in BSA blocking solution for 1 h. Tissues were then incubated with primary antibody at 4 °C overnight, followed by a rinse, then incubated with secondary antibody for 1 h. The tissue was then place in SBB solution for 20 min followed by a rinse in 70% ethanol and rinse in PBS. Nuclear stain (DAPI) was added, the slide was rinsed, and a cover slip was added. Slides were imaged with fluorescent microscopy at 40x using the appropriate excitation/emission filter and digitally recorded. For quantification of expression, RGB overlay signals were split and analyzed for specific fluorescence using image densitometry with Image J software (NIH, Bethesda, MD, USA). SBB, primary antibody, and secondary antibody were used as controls (data not shown). A minimum of 4–5 locations on intestinal villi for each section with 5 sections on each slide and *n* = 3–6 per group were imaged analyzed. Expression of tight junctions (claudin-3, occludin, and ZO-1) were measured by tracing specifically around enterocytes in villi. For all analyses, luminal contents were excluded. 

### 2.6. Real-Time RT-qPCR

mRNA expression for TJ and inflammatory markers in the small intestine were determined by real-time RT-qPCR (*n* = 8). RNA extraction from combined duodenum, jejunum, and ileal portions (equal portions) was performed using RNAEasy Mini kit (Qiagen, Valencia, CA, USA) per manufacturers protocol, and cDNA was synthesized using iScript cDNA Synthesis kit (Biorad, Hercules, CA, USA, cat. #170-8891). Analysis of claudin-3, occludin, ZO-1, TNF-α, MUC2, MMP-9, and IL-10 was conducted using specific primers (Table 1) and SYBR green detection (Sso Advanced Universal SYBR Green Supermix, Biorad), following manufacturer protocol, and were processed on Biorad CFX96, and ΔΔC_T_ values were calculated and normalized to GAPDH, as previously described by our laboratory [39].

### 2.7. Statistical Analysis

Data were analyzed by two-way ANOVA with Holm–Sidak multiple comparison all-pairwise test using GraphPad Prism 9 (Graphpad Software, Inc., San Diego, CA, USA). For regional small intestine immunofluorescent expression, a two-way ANOVA was performed using quantified expression from the duodenum, jejunum, and ileum. For global small intestine immunofluorescent expression, a two-way ANOVA was performed using quantified expression from combined regional expression. Intestinal immunofluorescence and RT-qPCR data are expressed as mean ± SEM and a *p* < 0.05 was considered statistically significant.

## 3. Results

### 3.1. Inhaled DEP Promotes Goblet Cell Formation in the Small Intestines

AB/PAS staining was used to determine mucus production and quantified goblet cells in the duodenum, ileum, and jejunum. Alcian blue stains highly acidic mucopolysaccharides blue and PAS stain neutral-acidic mucopolysaccharides pink. Intestinal goblet cells contain both neutral and acidic mucins which can be determined by a deep purple stain within the intestinal villi structure. Figure 1A–L shows representative images of AB/PAS staining through the regions of the small intestine and white arrows identify goblet cells within the villi. We observed a regional difference with DEP and probiotic treatment in the duodenum and jejunum when compared to CON (Figure 1M); however, there was no change in mucus production across any of the regions of the small intestines (Figure 1N). When looking at global expression throughout all regions of the intestine combined, we observed a significant increase in goblet cell formation with DEP exposure compared to CON (*p* = 0.049) (Figure 1O); however, there were no differences with exposure or probiotics in mucus production in the small intestines (Figure 1P). 

### 3.2. Inhaled DEP Increases MUC2 Expression Regionally and Globally in the Small Intestine, Regardless of Probiotic Treatment

To determine the expression and localization of MUC2 protein within the small intestine, we used immunofluorescent staining in the duodenum, jejunum, and ileum. Figure 2A–L shows representative images of MUC2 (green) expression within the three regions of the small intestine for all exposure and probiotic groups. Quantification of MUC2 showed increased expression with DEP exposure in duodenum, jejunum, and ileum (Figure 2M) and globally (Figure 2N). Interestingly, we also observed a significant increase in MUC2 expression in CON + PRO compared to CON, both globally and within the jejunum. The statistical values for MUC2 protein expression in the intestine were exposure F = 56.95, *p* < 0.001; probiotic treatment F = 12.45, *p* < 0.001; exposure x probiotics F = 0.359, *p* = 0.550. We also quantified MUC2 transcription expression in the small intestine by RT-qPCR. Unlike that observed at the protein level, there were no differences noted in MUC2 mRNA transcript across any of the exposure or probiotic treatment groups (Figure 2O). 

### 3.3. Inhaled DEP Results in Altered Claudin-3 Expression in the Intestine

To determine the expression and localization of TJ protein claudin-3 within the small intestine, we used immunofluorescent staining in the duodenum, jejunum, and ileum. Figure 3A–L shows representative pictures of claudin-3 (red) expression throughout the small intestine in all four groups. We found a significant increase in the jejunum (Figure 3M) in DEP compared to the CON group which accounted for the majority of the global (Figure 3N) statistically significant increase we observed in the small intestine. We also found that probiotic treatment did not mitigate the increase in claudin-3 expression with DEP exposure globally (Figure 3N) but did mitigate the response within the jejunum (Figure 3M). Interestingly, we observed a significant decrease in claudin-3 in CON + PRO compared to the CON group globally (Figure 3N) and within the duodenum and ileum (Figure 3M). Although we did not observe a statistically significant increase in claudin-3 in DEP + PRO compared to CON, it is important to note that the probiotics control group (CON + PRO) has a reduced baseline claudin-3 expression. We found overall claudin-3 protein expression was altered by DEP exposure and probiotic treatment (exposure F = 26.18, *p* < 0.001; probiotic F = 25.29, *p* < 0.001; exposure x probiotic F = 0.001, *p* = 0.972). We also quantified claudin-3 mRNA expression within the small intestine by RT-qPCR and found no statistical differences across any of the exposure or probiotic treatment groups at the transcript level (Figure 3O). 

### 3.4. Inhaled DEP Results in Increased Occludin Expression in the Small Intestine

To determine the expression and localization of tight junction occludin protein within the small intestine, we used immunofluorescent staining in the duodenum, jejunum, and ileum. Figure 4A–L shows representative images of occludin (red) expression throughout the regions of the small intestine for all groups. We observed a significant increase in occludin within the duodenum in DEP + PRO when compared to CON, DEP, and CON + PRO groups (Figure 4M), which accounts for the significant increase in DEP + PRO we observed globally (Figure 4N). Furthermore, we found a significant increase in occludin with probiotic treated groups in jejunum and ileum. While at a regional level, there is no statistically significant increase in occludin in DEP compared to the CON group (Figure 4N), we found a significant increase in occludin globally (Figure 4M). The two-way ANOVA shows occludin expression was driven by both DEP exposure and probiotic treatment (exposure F = 8.506, *p* = 0.004; probiotics F = 41.73, *p* < 0.001; exposure x probiotics F = 1.8, *p* = 0.997). We also measured occludin mRNA transcript expression in the small intestine by RT-qPCR (Figure 4O) and observed a significant decrease with CON + PRO and a noteworthy decrease with DEP when compared to CON. 

### 3.5. Inhaled DEP and Probiotic Treatment Alter ZO-1 Expression Regionally and Globally throughout the Small Intestine

To determine the expression and localization of intracellular TJ protein ZO-1 within the small intestine, we used immunofluorescent staining in the duodenum, jejunum, and ileum. Figure 5A–L shows representative pictures of ZO-1 (red) expression throughout the regions of the small intestine for all groups. Globally within the small intestine, we found a significant increase in ZO-1 expression with DEP compared to CON (Figure 5N), which expressed significantly higher in the duodenum (Figure 5M). Interestingly, we found that probiotics resulted in an increase in ZO-1 expression, which was observed regionally and globally. When compared to CON + PRO, ZO-1 expression in CON + DEP was significantly lower in the ileum (Figure 5M) only, which accounts for the overall global decrease in ZO-1 (Figure 5N). Overall, ZO-1 expression in the small intestine was significantly altered by probiotics and exposure-probiotic interactions (exposure F = 2.579, *p* = 0.110; probiotics F = 11.09 *p* = 0.001; exposure × probiotics F = 18.71, *p* < 0.001). We also measured the expression of ZO-1 mRNA and found when compared to CON, there was a significant decrease in DEP and CON + PRO groups (Figure 5O). Furthermore, there was a significant increase in ZO-1 in DEP + PRO when compared to the CON + PRO group. 

### 3.6. Inhaled DEP Results in an Increased Regional and Global Expression of MMP-9 in the Small Intestine 

Since MMP-9 is associated with altered TJ protein expression and permeability in the small intestine, we analyzed its expression via immunofluorescent staining of the duodenum, jejunum, and ileum. Figure 6A–L shows the representative images for MMP-9 (red) expression throughout the regions of the small intestine for all groups. We observed a significant increase in MMP-9 expression in the duodenum and jejunum (Figure 6M) and globally (Figure 6N) with DEP compared to CON. We observed no significant change for MMP-9 regionally or globally for DEP + PRO compared to CON + PRO (Figure 6M,N). We see an increased expression of MMP-9 in CON + PRO when compared to CON (Figure 6N). Intestinal MMP-9 expression was driven by DEP exposure and probiotics (exposure F = 17.28, *p* < 0.001; probiotics F = 0.328, *p* = 0.568; exposure x probiotics F = 21.79, *p* < 0.001). We also quantified MMP-9 mRNA transcript expression in the small intestine and observed no significant differences (Figure 6O). 

### 3.7. Probiotic Treatment Promotes Intestinal Inflammatory Response When Exposed to Inhaled Diesel Exhaust Particles

To determine whether DEP exposure promoted inflammation, we analyzed expression and localization of inflammatory markers TNF-α and IL-10 via double-immunofluorescent staining in the duodenum, jejunum, and ileum. Figure 7A–L shows representative images of TNF-α (red), IL-10 (green), and co-localization (yellow) throughout the regions of the small intestine for all groups. For TNF-α we observed a significant decrease in the duodenum and jejunum regions (Figure 7M) and globally (Figure 7N) in DEP compared to the CON group. Furthermore, we found the opposite response in probiotic groups where TNF-α was significantly increased in the jejunum and ileum regions (Figure 7M) and globally (Figure 7N) in DEP + PRO compared to CON + PRO. We saw similar trends with IL-10 (Figure 7O,P). We observed a large significant decrease in IL-10 in duodenum and jejunum regions (Figure 7O) and globally (Figure 7P) when comparing DEP to CON group. However, we see opposite results with probiotics, where we observed a significant increase in IL-10 in the jejunum (Figure 7O) and globally (Figure 7P) when comparing DEP + PRO to CON + PRO group. We found a decrease in inflammatory factors with probiotics, for both TNF-α and IL-10 we observed a regional and global significant decrease when comparing CON + PRO to CON. Statistical analyses showed significant alterations with DEP exposure and probiotics for TNF-α (exposure F = 0.687, *p* = 0.408; probiotics F = 0.444, *p* = 0.506; exposure x probiotics F = 43.15, *p* < 0.001) and IL-10 (exposure F = 7.32, *p* = 0.007; probiotics F = 0.004, *p* = 0.948; exposure × probiotics F = 38.47, *p* < 0.001).

### 3.8. Inhaled DEP Stimulates TLR-4 Expression with No Associated Effects on NF-κB in the Small Intestine

To better understand the involvement with TLR-4 and NF-κB in intestinal barrier integrity, we used immunofluorescence to quantify protein expression throughout the duodenum, jejunum, and ileum. Figure 8A–L shows represented pictures of TLR-4 (green), NF-κB (red), and co-localization (yellow) in the regions within the small intestine for each group. When analyzing NF-κB, we found a significant decrease in the jejunum but a significant increase in the ileum (Figure 8M) and no change globally (Figure 8N) when comparing DEP to CON. We did, however, find that probiotics resulted in a significant increase in NF-κB in the ileum (Figure 8M) and globally (Figure 8N) when comparing CON + PRO and DEP + PRO to CON. We found that TLR-4 was significantly increased with DEP compared to the CON group in all regions and globally (Figure 8O,P). Interestingly, we found that TLR-4 was significantly decreased in the jejunum and globally with DEP + PRO compared to CON + PRO. As we observed with inflammatory markers above, we found that TLR-4 was significantly increased in CON + PRO compared to CON. Statistical analyses showed that TLR-4 was significantly affected by both exposure and probiotics (exposure F = 0.286, *p* = 0.593; probiotics F = 18.38, *p* < 0.001; exposure x probiotics F = 44.08, *p* = 0.001), yet NF-κB was only affected by probiotics (exposure F = 1.17, *p* = 0.280, probiotics F = 13.76, *p* < 0.001, exposure × probiotics F = 0.246, *p* = 0.620).

## 4. Discussion

Inhalation exposure to traffic-generated PM has been linked with multiple disease states, including gastrointestinal disease, with a 40% increase in IBD hospitalizations [7]. While the mechanisms involved are not fully understood, air pollutants are thought to drive gut dysbiosis and systemic inflammation, which alters the epithelial barrier both through direct and indirect effects. Additionally, a HF diet consumption is understood to increase baseline inflammation both systemically and locally in the intestines [9,40]. Probiotics have shown promising results in promoting a healthy gut microbiome, intestinal integrity, and immunomodulation, yet probiotics have not yet been investigated as a possible treatment for gut permeability caused by environmental stressors [25,41]. In this study, we show for the first time to our knowledge that exposure to inhaled DEP in conjunction with a HF-diet ± probiotic-treatment results in regional and global alterations in the integrity of the small in C57Bl/6 male mice (Figure 9).

The mucosal layer is vital in gut homeostasis by providing defense against pathogen infiltration, limiting contact with ingested or microbial-derived toxins, and allowing beneficial microbes to adhere and colonize [42]. We found no mucus degradation or overproduction with DEP exposure ± probiotics; however, we observed an increase in goblet cell production in the intestines of the DEP-exposed animals compared to their respective controls. Additionally, we quantified the expression of MUC2, a major mucin present in the intestine, to determine if the increase in goblet cells was correlative to mucin production. We observed a global and regional increase in MUC2 protein expression with DEP exposure ± probiotics. From these same study animals, we previously reported an increase in *Akkermansia* with DEP exposure ± probiotics [37]. *Akkermansia* is a novel bacteria known for its mucin degrading capabilities and importance in mucosal integrity homeostasis [43]. As such, it is plausible that DEP exposure results in expansion of mucin-degrading *Akkermansia*, which initiates an increase in goblet cell formation and subsequent MUC2 release as positive feedback to the mucus degradation; however, further studies are necessary to determine the mechanisms involved.

To assess TJ protein expression, we chose three that are abundantly expressed throughout all regions of the small intestine. Somewhat surprisingly, we found a global increase in the protein expression of claudin-3, occludin, and ZO-1 with DEP exposure. The observed increase in TJ protein expression may be in response to intestinal injury occurring at an earlier time point in the DEP exposure; however, analyses of additional exposure time points are necessary. At the transcript level, we found an inverse correlation for occludin, and ZO-1 expression in the intestines, compared to the protein level. While mRNA and protein expression are not always mutually exclusive, especially during stress responses, it is plausible that post-translational modifications or stress-induced responses are occurring that down-regulate transcription [44,45]. Furthermore, the mice in the current study were sacrificed 24 h following the last DEP exposure, which could promote an acute phase stress response leading to decreased transcription factors in these animals [46]. In our previous preliminary study identifying MVE and WS effects on gut integrity, we also observed a significant decrease in claudin-3 protein in the small intestine with exposure; however, contrary to the current study, we observed a decrease in occludin protein levels with exposure [34]. However, this previous study was a 50-d exposure study with different exposure methodology and concentrations and only focused on endpoints within the duodenum and ileum. While a follow-up dose-time dependent study is warranted, we propose that DEP is driving the upregulation of TJ proteins as a protective mechanism, at this concentration and time point. 

We also observed a variation in the expression of TJs in different regions throughout the intestines of both the CON and DEP exposed animals in PRO groups, suggesting that the bacterial species in the probiotics are influencing the varying expression of TJs. We see a decrease in claudin-3 protein expression in the intestines, both regionally and globally, in the CON + PRO animals compared to CON animals. It has been reported that claudin expression in intestinal disease states changes depending on the localization and expression of other claudins [47]. Conflicting studies have shown that claudin-3 reduction is associated with no change in claudin-4 [48] and that claudin-3 remains stable while claudin-4 expression is reduced [49]. These studies suggest that claudin expression in the intestines is, to some extent, regulated by other claudins. Thus, we propose that the decrease in claudin-3 observed in the CON + PRO group is associated with the possible increase in other claudin TJs. Furthermore, ZO-1 and ZO-2 were shown to regulate localization and initiation of claudin polymerization in epithelial cells [50]. Considering the significant decrease in claudin-3 and a significant increase in ZO-1 expression in CON + PRO groups, we propose that ZO-1 is mediating these responses in claudin expression. When comparing the expression of TJ proteins in the DEP + PRO group with CON + PRO, we see the same trend with exposure on claudin-3 and occludin expression as that in the DEP exposed animals that did not receive probiotics; however, we see a decrease with global ZO-1 expression that was only observed regionally in the ileum. While no studies have yet determined the effects of inhaled particulate matter on ZO-1 and the microbiome, a study by Mutlu et al. showed that ingested PM resulted in increased intestinal permeabilities characterized by decreased ZO-1 transcript expression in C57Bl/6 mice on a regular chow diet [51]. In addition to the previously mentioned relationship between ZO-1 and claudin expression, we suspect that the decrease in ZO-1 in DEP + PRO is perhaps due to ingested particles following mucociliary clearance. We reported that DEP exposure increased ZO-1, yet an opposite effect was seen with probiotics where ZO-1 decreased with DEP exposure compared to their respective controls. However, when comparing both DEP exposures, we find that there is no significant difference. Nascimento et al. recently reported that a HF diet resulted in decreased ZO-1 and claudin-3 in the duodenum and jejunum after 30 d compared to low-fat diet control in wild-type mice [52]. Furthermore, probiotic treatment has been shown to promote intestinal integrity by strengthening TJs in the intestine [53,54,55]. We propose that the HF diet in the CON group is decreasing baseline ZO-1 expression, which is then normalized by probiotics, thus resulting in the inverse correlation with DEP exposure between probiotics and animals not treated with probiotics. 

MMP-9 is a pleiotropic enzyme that responds to tissue damage and when activated contributes to tissue remodeling. Several studies in the literature report a protective role of MMP- during periods of stress [56,57,58]. For example, a recent study reported that MMP-9 was found to reduce reactive oxygen species and DNA damage in colitis-associated cancer, suggesting its protective role in the intestine [57]. Another study found that the fecal microbiome of MMP-9^−/−^ mice showed an expansion of Proteobacteria compared to wild-type mice, providing evidence that MMP-9 expression is correlative to the presence of certain microbial profiles. Interestingly, we previously reported expansion of Proteobacteria with DEP exposure in the intestines from these same study mice, yet we observe an increase in MMP-9 protein expression throughout the duodenum and jejunum [37]. We see the highest expression of MMP-9 in the duodenum of DEP-exposed animals, which could result from the ingestion of particles encountering the duodenum at the highest concentration before dispersing as luminal contents move throughout the small intestine. Furthermore, we see attenuation of MMP-9 expression with probiotic treatment in the intestines DEP exposed animals. While there is little information regarding probiotic treatment and MMP-9 expression in the intestines, a study by Garg et al. found that MMP-9 regulated goblet cell differentiation and MUC2 expression in the colon of MMP-9^−/−^ mice [59]. We found that our probiotic control group had increased MUC2 expression and increased MMP-9 expression when compared to CON. Although further studies are necessary, this correlation suggests that MMP-9 expression plays a role in the protective responses we see in these mice. 

We quantified the expression of pro-inflammatory cytokine TNF-α in the intestines of our study animals to determine if DEP exposure promoted a local inflammatory response. Unexpectedly, we found that DEP exposure resulted in an overall decrease in the expression of TNF-α in the intestines compared to CON animals. A HF diet is known to contribute to both systemic and intestinal inflammation, and several studies have shown increased TNF-α expression in the intestine with a HF diet in mice [60,61]. TNF-α is multifunctional in that it promotes cell survival by NF-κB signaling activation or initiates programmed cell death; as such, TNF-α plays a prominent role in the homeostatic functions of the gut [62]. Moreover, anti-TNF-α treatments in humans and mice have been associated with the expansion of *Escherichia* in the microbiome, suggesting a correlation between *Escherichia* and reduction in TNF-α [63,64]. Thus, we propose that the HF diet is likely contributing to the increased baseline expression of TNF-α in the intestines of the CON group in this study. Moreover, we suggest that the reduction in TNF-α with DEP exposure is correlated to the observed expansion in *Escherichia*, which we previously reported [37]. When comparing PRO + CON to CON, we find a significant reduction in TNF-α expression, which we attribute to the known anti-inflammatory effects of probiotics, namely *Lactococcus* and *Lactobacillus* strains [65,66]. Furthermore, we observe a significant increase in TNF-α with DEP + PRO compared to its respective CON + PRO control, suggesting that the immunomodulatory effects of probiotics normalized the immune response in the intestines which resulted in an augmented response with DEP exposure. We have previously reported that probiotic treatment mitigates the expansion of *Escherichia* observed with DEP exposure in the HF diet mice, confirming the possible role of *Escherichia* in TNF-α signaling in these mice [37]. 

We also measured anti-inflammatory cytokine IL-10 expression in the intestines of our study mice, and we observed the same trend as TNF-α. IL-10 is produced by nearly all cell types and is an important regulator of innate and adaptive immune responses in the intestine [67,68]. We found that DEP exposure resulted in a reduction in IL-10 expression when compared to CON. A study by Kish et al. showed that ingested environmental PM increase IL-10 after 7 d of exposure, which was resolved by exposure day 14 in the small intestine in wild-type mice [69]. While our study is an inhalational exposure, we assume that at least some of the DEP are likely ingested via mucociliary clearance. Therefore, it is plausible that an acute response promotes an initial increase in IL-10 in the intestines, but by the 30-d time point of the current study, there is a depletion of IL-10. Notably, we see an opposite response in IL-10 in our probiotic treatment groups, where IL-10 is increased in DEP + PRO compared to CON + PRO. Multiple studies have shown that probiotics consisting of different species of *Bifidobacterium, Lactobacillus,* and *Lactococcus* increase the expression of IL-10 in the intestines [68,70,71]. Additionally, such probiotics have been shown to be effective in mitigating HF diet responses in the intestines by increasing antioxidants, reducing low-density lipoprotein (LDL) cholesterol levels, regulating lipid metabolism hormones, and mediating signaling pathways [72,73,74,75]. A study by Holowacz et al. determined that a HF diet resulted in a significant increase in cytokine CCL-2 in the intestine and proinflammatory mediator leukotriene in the intestines and adipose tissue, which was normalized back to low fat-fed animal measurements by probiotic treatment with *Lactobacillus* and *Bifidobacterium* species [76]. We previously showed that *Lactococcus* increased in both probiotic treated groups compared to those not treated with probiotics and that DEP exposure resulted in a reduction of *Bifidobacterium,* which was attenuated with probiotic treatment [37]. Considering the known effects of probiotics containing *Bifidobacterium* in a HF diet and *Lactobacillus* in intestinal inflammation, it is plausible that probiotic treatment in this study functions as immunomodulators and serves to normalize HF diet-mediated alterations in inflammatory responses. 

It is also important to note the apparent regional differences in inflammatory responses in intestines of our study animals across exposure and probiotic-treatment groups. We found that the jejunum resulted in the most variation in immune response for exposure and treatment. This is important in future applications of intestinal research to consider the dynamic and varying responses throughout the entire small intestine. We have provided a summary of the regional and global findings for all endpoints when comparing DEP to CON (Table 2), CON + PRO to CON (Table 3), CON + DEP vs. CON + PRO (Table 4), and comparing DEP + PRO to DEP (Table 5). 

NF-κB activity is responsible for innate and adaptive immune response and is of particular importance in the maintenance of the intestinal epithelial barrier [77]. While we did observe a regional variation of NF-κB in the intestine (e.g., no change in the duodenum, decreased in the jejunum, increased in the ileum), we found no change in NF-κB expression with DEP exposure when the regions were averaged for global expression across the intestines, compared to CON. However, probiotic treatment significantly increased NF-κB expression in the ileum in both the CON and DEP groups, leading to an overall increase in averaged global expression. Probiotics are known for their ability to stimulate an immune response and studies have shown that *Lactobacillus* probiotics can initiate immunity through NF-κB signaling pathways [78,79]. Due to the varying immune responses between our probiotic and non-probiotic groups, probiotics in this study may be mediating an immune response via NF-κB. 

Activation of TLRs are known to promote NF-κB signaling pathways. We measured TLR-4 in the small intestine and found that DEP exposure resulted in an increase in TLR-4 expression across all regions of the small intestine analyzed. We have previously reported an increase in *Escherichia* in the small intestines of this group. This increase of gram-negative bacteria in the gut may account for the upregulation in TLR-4 expression. Interestingly, we found that probiotic treatment also resulted in an increase in TLR-4 expression when comparing CON + PRO to CON. Moreover, we see a significant decrease in TLR-4 with DEP + PRO compared to CON + PRO. Probiotics are known to modulate TLR expression, which has been shown to protect against pathogens [80]. We propose that probiotics stimulate upregulation of TLR-4 expression, which is reduced with DEP exposure. This correlates with the increased TNF-α expression we see in DEP + PRO compared to CON + PRO, as TLR-4 induces TNF-α expression. We did not find the same trend in the expression of TLR-4 and NF-κB, which indicates that TLR-4 is not entirely responsible for the expression of NF-κB in the intestines of these mice.

Although we observed alterations in the intestine at this DEP exposure concentration and 30-d time point, these responses likely occur in dose and time dependent phases. Analysis of these same endpoints in the intestines at different exposure concentrations and/or durations would likely yield varied results, especially considering the variability and dynamic environment of the intestine, which could be viewed as a limitation of the current study. We chose oropharyngeal aspiration (OA) exposure to DEP instead of whole-body inhalation to eliminate ingestion exposure via oral cavity, grooming, or food. This exposure approach allowed for specifically investigating the effects of inhaled PM from the lungs; however, it is also a noted limitation of the study in so much that OA exposure is not consistent with nasal inhalation more relevant to human exposure scenarios. Additionally, we chose to provide probiotics via drinking water in this study to limit the stress of daily oral gavage to mice. Thus, we could only measure the average consumption of probiotics via drinking water per animal/per cage each day. The authors also note that all of the study animals were on a HF diet, without a low-fat diet control for comparison, which limits the understanding of the outcomes in the intestines resulting from diet vs. exposure, and how these may be impacted through probiotic use. Nonetheless, this study has provided novel information on the effects of DEP exposure, when combined with a HF diet, on the integrity of different regions of the small intestine. Furthermore, to our knowledge, this is the first study that assessed the outcomes of probiotic treatment coupled with traffic-generated PM exposures on the intestines. The current study findings also highlight the importance of investigating changes in the intestines regionally and globally to determine responses in the small intestine. We observed several instances where regional differences in protein or gene expression may not account for the global small intestine response, and thus this is important to consider in future intestinal and microbiome research. 

This study aimed to determine if probiotic treatment could “protect” the gut microbiome from DEP exposure-mediated alterations in the intestines. Our results revealed that the microbiome plays a vital role in mediating these systemic inflammatory responses following a 30-d DEP exposure and a HF diet protocol. The amount of clearance of the DEP from the lungs and subsequent ingestion via the mucociliary tract is still not fully understood, so we can only speculate on the direct and indirect influence of the DEP particles on gut bacteria. Previous ambient PM and vehicle exhaust exposure studies have reported alterations in the gut microbiome profile and systemic inflammation [33,34]; however, the effects of traffic-generated PM, such as DEP, on these outcomes is not as well characterized. When adjusted for mice, the average DEP dosage of ~10 µg/mouse daily is expected to be 40-fold higher than the comparable alveolar deposition from a 24-h inhalation of 100 µg/m^3^ in humans [79]. The dose of DEP used in the current study may be higher than that typically experienced in an ambient environmental scenario; however, we chose this dose based on inflammatory outcomes and alterations in microbiota profiles previously described in the lung and cardiovascular system of wildtype mice [36,37,81]. In doing so, it allowed for comparisons of outcomes described at the molecular and tissue levels in the gut to those described systemically and within the cardiopulmonary system. We also recognize that the DEP utilized for this study is not necessarily representative of all of the diesel engine-generated PM. Nevertheless, this study provides preliminary information on the outcomes of subacute DEP exposure and diet-mediated alterations on gut microbial profiles and associated systemic inflammatory signaling pathways.

## 5. Conclusions

Our findings demonstrate that inhaled DEP exposure alters intestinal integrity and inflammation in conjunction with a HF diet (Figure 9). The use of probiotics in this study proved to be fundamental in the understanding of the influence the microbiome has on the regulation of intestinal integrity and inflammation during inhaled DEP exposure. Additionally, this study serves as a foundation for future studies of regional vs. global expression in the small intestines and the potential use of probiotics in environmental exposure research. 

## Figures and Tables

**Figure 1 cells-11-01445-f001:**
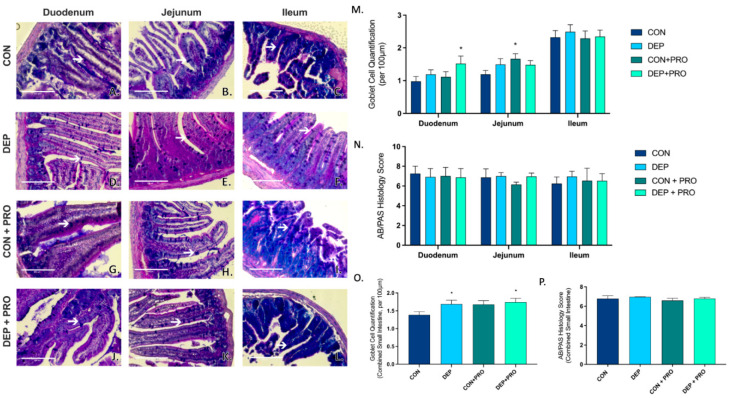
Exposure to inhaled diesel exhaust particles promotes goblet cell formation in the small intestine of C57Bl/6 male mice. Representative images of AB/PAS staining in the three regions of the small intestine of C57Bl/6 male mice on a high-fat diet exposed to saline (CON; **A**–**C**), diesel exhaust particles (DEP-35 µg PM; **D**–**F**), saline and probiotics (CON + PRO, 0.3 g/d of Ecologic^®^ Barrier probiotics; **G**–**I**), or diesel exhaust particles and probiotics (DEP + PRO, 0.3 g/d of Ecologic^®^ Barrier probiotics; **J**–**L**) twice a week for 4 w. Panels show representative images within the duodenum (**A**,**D**,**G**,**J**), jejunum (**B**,**E**,**H**,**K**), and ileum (**C**,**F**,**I**,**L**). Graph (**M**) shows the quantification of goblet cells per 100 µm of intestinal villi by region and (**O**) shows the global (cumulative quantification of three portions) quantification of goblet cells per 100 µm of intestinal villi in the small intestine. White arrow indicates goblet cell. Graph (**N**) shows the regional histological mucus score, and (**P**) shows the global (cumulative) histological mucus score. 40x magnification, scale bar = 100 µm. Data are depicted as ± SEM with * *p* < 0.05 compared to CON.

**Figure 2 cells-11-01445-f002:**
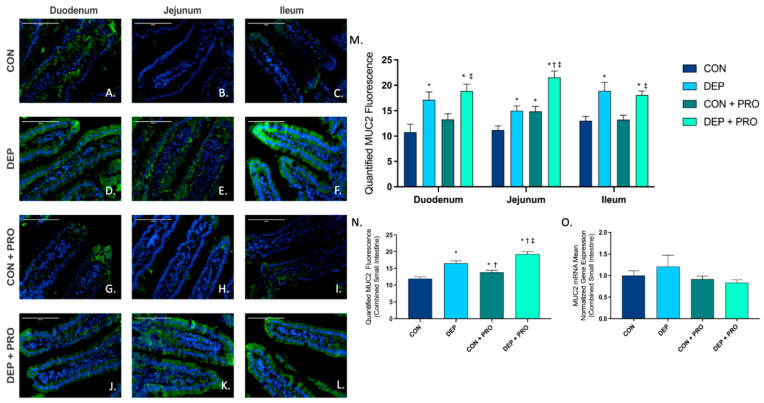
Exposure to inhaled diesel exhaust particles increases mucin 2 (MUC2) expression in the small intestine of C57Bl/6 male mice, regardless of probiotic treatment. Representative images of MUC2 expression in the three regions of the small intestine of C57Bl/6 male mice on a high-fat diet exposed to saline (CON; **A**–**C**), diesel exhaust particles (DEP-35 µg PM; **D**–**F**), saline and probiotics (CON + PRO, 0.3 g/d of Ecologic^®^ Barrier probiotics; **G**–**I**), or diesel exhaust particles and probiotics (DEP + PRO, 0.3 g/d of Ecologic^®^ Barrier probiotics; **J**–**L**) twice a week for 4 w. Panels show merged images within the duodenum (**A**,**D**,**G**,**J**), jejunum (**B**,**E**,**H**,**K**), and ileum (**C**,**F**,**I**,**L**). Green fluorescence indicates MUC2 expression and blue fluorescence indicates nuclear staining (Hoechst). Graph (**M**) shows the histological analysis of MUC2 by region, (**N**) shows the global (cumulative expression of three portions) analysis of MUC2 expression in the small intestine, and (**O**) shows the global mean normalized gene expression of MUC2 mRNA transcript expression in the small intestine, as determined by RT-qPCR. 40× magnification, scale bar = 100 µm. Data are depicted as ± SEM with * *p* < 0.05 compared to CON, † *p* < 0.05 compared to DEP, and ‡ *p* < 0.05 compared to CON + PRO.

**Figure 3 cells-11-01445-f003:**
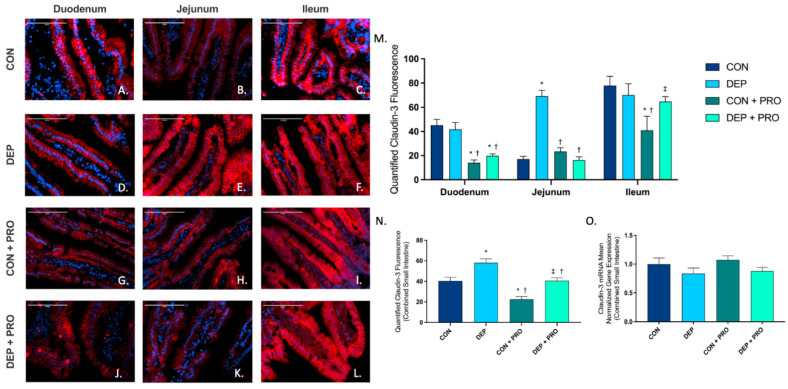
Exposure to inhaled diesel exhaust particles results in an increase in claudin-3 expression in the small intestine of C57Bl/6 male mice. Representative images of claudin-3 expression in the three regions of the small intestine of C57Bl/6 male mice on a high-fat diet exposed to saline (CON; **A**–**C**), diesel exhaust particles (DEP-35 µg PM; **D**–**F**), saline and probiotics (CON + PRO, 0.3 g/d of Ecologic^®^ Barrier probiotics; **G**–**I**), or diesel exhaust particles and probiotics (DEP + PRO, 0.3 g/d of Ecologic^®^ Barrier probiotics; **J**–**L**) twice a week for 4 w. Panels show merged images within the duodenum (**A**,**D**,**G**,**J**), jejunum (**B**,**E**,**H**,**K**), and ileum (**C**,**F**,**I**,**L**). Red fluorescence indicates claudin-3 expression and blue fluorescence indicates nuclear staining (Hoechst). Graph (**M**) shows the histological analysis of claudin-3 by region, (**N**) shows the global (cumulative expression of three portions) analysis of claudin-3 expression in the small intestine, and (**O**) shows the global mean normalized gene expression of claudin-3 mRNA transcript expression in the small intestine, as determined by RT-qPCR. 40× magnification, scale bar = 100 um. Data are depicted as ± SEM with * *p* < 0.05 compared to CON, † *p* < 0.05 compared to DEP, and ‡ *p* < 0.05 compared to CON + PRO.

**Figure 4 cells-11-01445-f004:**
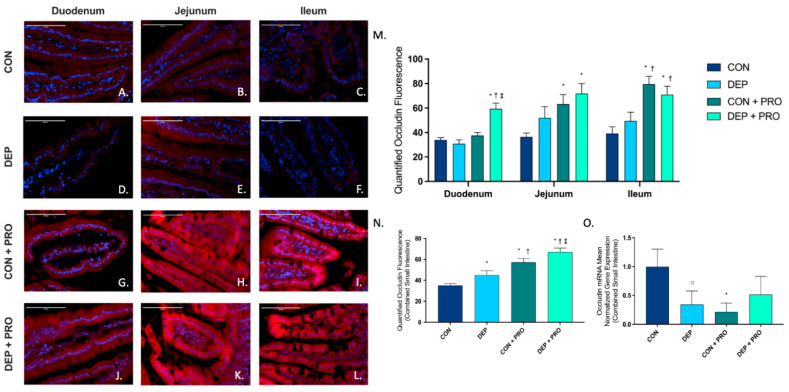
Exposure to inhaled diesel exhaust particles result in an increase in occludin expression in the small intestine of C57Bl/6 male mice. Representative images of occludin expression in the three regions of the small intestine of C57Bl/6 male mice on a high-fat diet exposed to saline (CON; **A**–**C**), diesel exhaust particles (DEP-35 µg PM; **D**–**F**), saline and probiotics (CON + PRO, 0.3 g/d of Ecologic^®^ Barrier probiotics; **G**–**I**), or diesel exhaust particles and probiotics (DEP + PRO, 0.3 g/d of Ecologic^®^ Barrier probiotics; **J**–**L**) twice a week for 4 w. Panels show merged images within the duodenum (**A**,**D**,**G**,**J**), jejunum (**B**,**E**,**H**,**K**), and ileum (**C**,**F**,**I**,**L**). Red fluorescence indicates occludin expression and blue fluorescence indicates nuclear staining (Hoechst). Graph (**M**) shows the histological analysis of occludin by region, (**N**) shows the global (cumulative expression of three portions) analysis of occludin expression in the small intestine, and (**O**) shows the global mean normalized gene expression of occludin mRNA transcript expression in the small intestine, as determined by RT-qPCR. 40× magnification, scale bar = 100 µm. Data are depicted as ± SEM with * *p* < 0.05 compared to CON, † *p* < 0.05 compared to DEP, ‡ *p* < 0.05 compared to CON + PRO, and ○ *p* < 0.07 compared to CON.

**Figure 5 cells-11-01445-f005:**
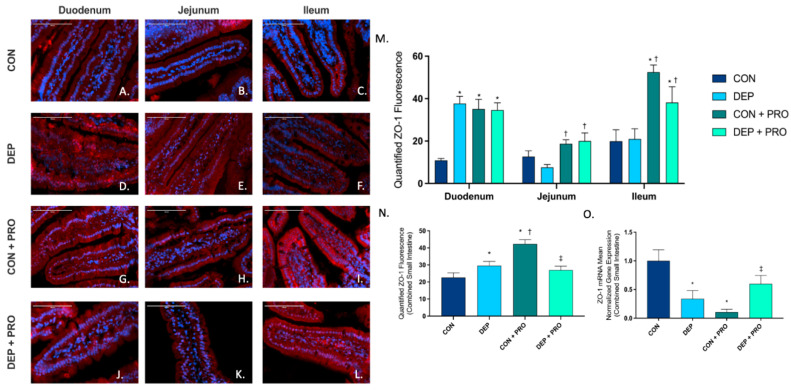
Exposure to inhaled diesel exhaust particles and probiotics alters zonula occludens 1 (ZO-1) expression regionally and globally within the small intestine of C57Bl/6 male mice. Representative images of ZO-1 expression in the three regions of the small intestine of C57Bl/6 male mice on a high-fat diet exposed to saline (CON; **A**–**C**), diesel exhaust particles (DEP-35 µg PM; **D**–**F**), saline and probiotics (CON + PRO, 0.3 g/d of Ecologic^®^ Barrier probiotics; **G**–**I**), or diesel exhaust particles and probiotics (DEP + PRO, 0.3 g/d of Ecologic^®^ Barrier probiotics; **J**–**L**) twice a week for 4 w. Panels show merged images within the duodenum (**A**,**D**,**G**,**J**), jejunum (**B**,**E**,**H**,**K**), and ileum (**C**,**F**,**I**,**L**). Red fluorescence indicates ZO-1 expression and blue fluorescence indicates nuclear staining (Hoechst). Graph (**M**) shows the histological analysis of ZO-1 by region, (**N**) shows the global (cumulative expression of three portions) analysis of ZO-1 expression in the small intestine, and (**O**) shows the global mean normalized gene expression of ZO-1 mRNA transcript expression in the small intestine, as determined by RT-qPCR. 40x magnification, scale bar = 100 µm. Data are depicted as ± SEM with * *p* < 0.05 compared to CON, † *p* < 0.05 compared to DEP, and ‡ *p* < 0.05 compared to CON + PRO.

**Figure 6 cells-11-01445-f006:**
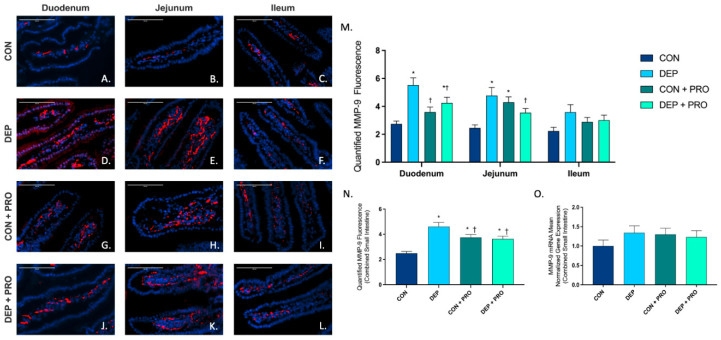
Exposure to inhaled diesel exhaust particles increases matrix metalloproteinase 9 (MMP-9) expression in the small intestine of C57Bl/6 male mice, which is not observed in probiotic treated mice. Representative images of MMP-9 expression in the three regions of the small intestine of C57Bl/6 male mice on a high-fat diet exposed to saline (CON; **A**–**C**), diesel exhaust particles (DEP-35 µg PM; **D**–**F**), saline, and probiotics (CON + PRO, 0.3 g/d of Ecologic^®^ Barrier probiotics; **G**–**I**), or diesel exhaust particles and probiotics (DEP + PRO, 0.3 g/d of Ecologic^®^ Barrier probiotics; **J**–**L**) twice a week for 4 w. Panels show merged images within the duodenum (**A**,**D**,**G**,**J**), jejunum (**B**,**E**,**H**,**K**), and ileum (**C**,**F**,**I**,**L**). Red fluorescence indicates MMP-9 expression and blue fluorescence indicates nuclear staining (Hoechst). Graph (**M**) shows the histological analysis of MMP-9 by region, (**N**) shows the global (cumulative expression of three portions) analysis of MMP-9 expression in the small intestine, and (**O**) shows the global mean normalized gene expression of MMP-9 mRNA transcript expression in the small intestine, as determined by RT-qPCR. 40× magnification, scale bar = 100 µm. Data are depicted as ± SEM with * *p* < 0.05 compared to CON, † *p* < 0.05 compared to DEP.

**Figure 7 cells-11-01445-f007:**
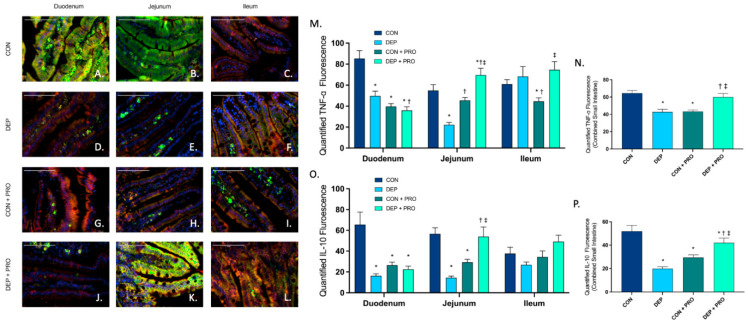
Probiotic intervention promotes intestinal inflammatory marker TNF- and IL-10 expression in the small intestine during exposure to inhaled diesel exhaust particles in C57Bl/6 male mice. Representative images of TNF-α and IL-10 expression in the three regions of the small intestine of C57Bl/6 male mice on a high-fat diet exposed to saline (CON; **A**–**C**), diesel exhaust particles (DEP-35 µg PM; **D**–**F**), saline and probiotics (CON + PRO, 0.3 g/d of Ecologic^®^ Barrier probiotics; **G**–**I**), or diesel exhaust particles and probiotics (DEP + PRO, 0.3 g/d of Ecologic^®^ Barrier probiotics; **J**–**L**) twice a week for 4 w. Panels show merged images within the duodenum (**A**,**D**,**G**,**J**), jejunum (**B**,**E**,**H**,**K**), and ileum (**C**,**F**,**I**,**L**). Red fluorescence indicates TNF-α expression, green fluorescence indicates iL-10 expression, yellow indicates co-localization of TNF-α and IL-10, and blue fluorescence indicates nuclear staining (Hoechst). Graph (**M**) shows the histological analysis of TNF-α by region, (**N**) shows the global (cumulative expression of three portions) analysis of TNF-α expression in the small intestine, (**O**) shows the histological analysis of IL-10 by region, (**P**) shows the global (cumulative expression of three portions) analysis of IL-10 expression in the small intestine 40× magnification, scale bar = 100 µm. Data are depicted as ± SEM with * *p* < 0.05 compared to CON, † *p* < 0.05 compared to DEP, and ‡ *p* < 0.05 compared to CON+PRO.

**Figure 8 cells-11-01445-f008:**
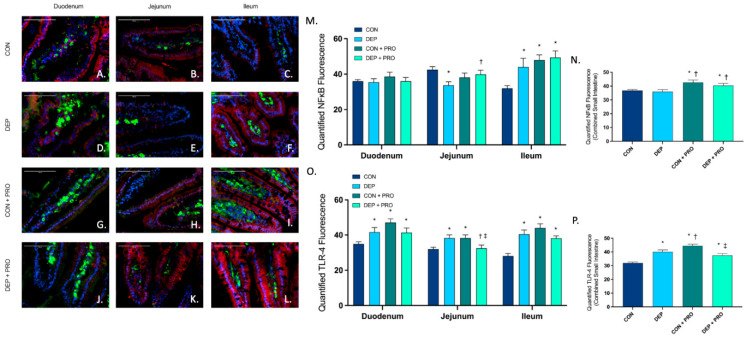
Exposure to inhaled diesel exhaust particles stimulates TLR-4 expression but not NF-κB in the small intestine of C57Bl/6 male mice. Representative images of NF-κB and TLR4 expression in the three regions of the small intestine of C57Bl/6 male mice on a high-fat diet exposed to saline (CON; **A**–**C**), diesel exhaust particles (DEP-35 µg PM; **D**–**F**), saline and probiotics (CON + PRO, 0.3 g/d of Ecologic^®^ Barrier probiotics; **G**–**I**), or diesel exhaust particles and probiotics (DEP + PRO, 0.3 g/d of Ecologic^®^ Barrier probiotics; **J**–**L**) twice a week for 4 w. Panels show merged images within the duodenum (**A**,**D**,**G**,**J**), jejunum (**B**,**E**,**H**,**K**), and ileum (**C**,**F**,**I**,**L**). Red fluorescence indicates NF-κB expression, green fluorescence indicates TLR-4 expression, yellow indicates co-localization of NF-κB and TLR-4, and blue fluorescence indicates nuclear staining (Hoechst). Graph (**M**) shows the histological analysis of NF-κB by region, (**N**) shows the global (cumulative expression of three portions) analysis of NF-κB expression in the small intestine, (**O**) shows the histological analysis of TLR-4 by region, (**P**) shows the global (cumulative expression of three portions) analysis of TLR-4 expression in the small intestine 40x magnification, scale bar = 100 µm. Data are depicted as ± SEM with * *p* < 0.05 compared to CON, † *p* < 0.05 compared to DEP, and ‡ *p* < 0.05 compared to CON + PRO.

**Figure 9 cells-11-01445-f009:**
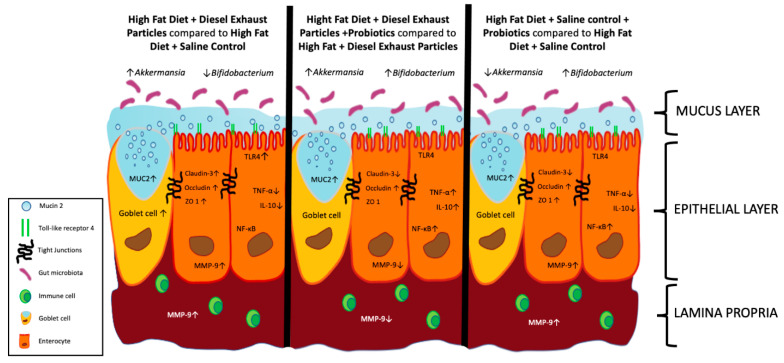
Summary of the effects of inhaled diesel exhaust particles (DEP) and probiotics (PRO) on intestinal integrity and associated microbiota. Exposure to inhaled DEP results increase (↑) Akkermansia and decreased (↓) Bifidobacterium associated with an increase in goblet cell formation and associated MUC2 expression, as well as an increase in claudin-3, occludin, ZO-1, MMP-9, and TLR-4, and a decrease in TNF-α and IL-10 when compared to saline control (CON). Probiotic treatment alters DEP exposure-response characterized by an increase (↑) in Akkermansia and Bifidobacterium and an increase in MUC2, occludin, TNF-α, IL-10, NFκB, and a decrease in claudin-3 and MMP-9. Probiotics without DEP exposure result in decreased (↓) Akkermansia and increase (↑) Bifidobacterium and an increase in MUC2, occludin, ZO-1, MMP-9, and NF-κB, and a decrease in claudin-3, TNF-α, and IL-10.

**Table 1 cells-11-01445-t001:** Primer sequences used for Real-time RT-qPCR.

Claudin-3	FP: 5′-CGTACCGTCACCACTACCAG RP: 5′-CTGTGTGTCGTCTGTCACCA
ZO-1	FP: 5′-TGGTCTGTTTGCCCACTGTT RP: 5′-TCTGTACATGCTGGCCAAGG
Occludin	FP: 5′-CCCTGACCACTATGAAACAG RP: 5′-TTGATCTGAAGTGATAGGTG
MUC2	FP: 5′-CCTGAAGACTGTCGTGCTGT RP: 5′-GGGTAGGGTCACCTCCATCT
MMP-9	FP: 5′-GACAGGCACTTCACCGGCTA RP: 5′-CCCGACACACAGTAAGCATTC
GAPDH	FP: 5′-CATGGCCTTCCGTGTTCCTA RP: 5′-GCGGCACGTCAGATCCA

Abbreviations: ZO-1, zonula occludens 1; TNF-α, tumor necrosis factor α; MUC2, mucin 2; MMP-9 matrix metalloproteinase 9; and IL-10, interleukin 10.

**Table 2 cells-11-01445-t002:** Summary of regional and global findings in the small intestine from DEP exposure (DEP vs. CON).

	Duodenum	Jejunum	Ileum	Global/Avg	Transcript mRNA Levels
Goblet Cell count	-	-	-	↑	N/A
MUC2	↑	↑	↑	↑	-
Claudin-3	-	↑	-	↑	-
Occludin	-	-	-	↑	-
ZO-1	↑	-	-	↑	↓
MMP-9	↑	↑	-	↑	-
TNF-α	↓	↓	-	↓	N/A
IL-10	↓	↓	-	↓	N/A
NF-κB	-	↓	↑	-	N/A
TLR-4	↑	↑	↑	↑	N/A

Abbreviations: MUC2, mucin 2; ZO-1, zonula occludens; MMP-9, matrix metalloproteinase 9; TNF-α, tumor necrosis factor-alpha; IL-10, interleukin 10; NF-κB, nuclear factor kappa B; and TLR-4, toll-like receptor 4. ↑ = increased expression, ↓ = decreased expression.

**Table 3 cells-11-01445-t003:** Summary of regional and global findings in the small intestine from probiotic treatment (CON + PRO vs. CON).

	Duodenum	Jejunum	Ileum	Global/Avg	Transcript mRNA Levels
Goblet Cell count	-	↑	-	-	N/A
MUC2	-	↑	-	↑	-
Claudin-3	↓	-	↓	↓	-
Occludin	-	↑	↑	↑	↓
ZO-1	↑	-	↑	↑	↓
MMP-9	-	↑	-	↑	-
TNF-α	↓	-	↓	↓	N/A
IL-10	↓	↓	-	↓	N/A
NF-κB	-	-	↑	↑	N/A
TLR-4	↑	↑	↑	↑	N/A

Abbreviations: MUC2, mucin 2; ZO-1, zonula occludens; MMP-9, matrix metalloproteinase 9; TNF-α, tumor necrosis factor-alpha; IL-10, interleukin 10; NF-κB, nuclear factor kappa B; and TLR-4, toll-like receptor 4. ↑ = increased expression, ↓ = decreased expression.

**Table 4 cells-11-01445-t004:** Summary of regional and global findings in the small intestine from DEP exposure and probiotic treatment (CON + DEP vs. CON + PRO).

	Duodenum	Jejunum	Ileum	Global/Avg	Transcript mRNA Levels
Goblet Cell count	-	-	-	-	N/A
MUC2	↑	↑	↑	↑	-
Claudin-3	-	-	↑	↑	-
Occludin	↑	-	-	↑	-
ZO-1	-	-	-	↓	↑
MMP-9	-	-	-	-	-
TNF-α	-	↑	↑	↑	N/A
IL-10	-	↑	-	↑	N/A
NF-κB	-	-	-	-	N/A
TLR-4	-	↓	-	↓	N/A

Abbreviations: MUC2, mucin 2; ZO-1, zonula occludens; MMP-9, matrix metalloproteinase 9; TNF-α, tumor necrosis factor-alpha; IL-10, interleukin 10; NF-κB, nuclear factor kappa B; and TLR-4, toll-like receptor 4. ↑ = increased expression, ↓ = decreased expression.

**Table 5 cells-11-01445-t005:** Summary of regional and global findings in the small intestine from probiotic treatment in DEP exposed animals (DEP + PRO vs. DEP).

	Duodenum	Jejunum	Ileum	Global/Avg	Transcript mRNA Levels
Goblet Cell count	-	-	-	-	N/A
MUC2	↑	↑	-	↑	-
Claudin-3	↓	↓	-	↓	-
Occludin	↑	-	↑	↑	-
ZO-1	-	↑	↑	-	-
MMP-9	↓	↓	-	↓	-
TNF-α	↓	↑	-	↑	N/A
IL-10	-	↑	-	↑	N/A
NF-κB	-	↑	-	↑	N/A
TLR-4	-	↓	-	-	N/A

Abbreviations: MUC2, mucin 2; ZO-1, zonula occludens; MMP-9, matrix metalloproteinase 9; TNF-α, tumor necrosis factor-alpha; IL-10, interleukin 10; NF-κB, nuclear factor kappa B; and TLR-4, toll-like receptor 4. ↑ = increased expression, ↓ = decreased expression.

## Data Availability

The datasets used and/or analyzed during the current study are available from the corresponding author upon reasonable request.

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
