# Peer review of "Probiotics Function as Immunomodulators in the Intestine in C57Bl/6 Male Mice Exposed to Inhaled Diesel Exhaust Particles on a High-Fat Diet"

_cells, 2022, doi:10.3390/cells11091445_

Round 1

Reviewer 1 Report

The study entitled “Role of inhaled diesel exhaust particles and probiotics in intestinal integrity and inflammation in C57Bl/6 male mice” by Phillippi et al. is well designed and performed, and results are clearly presented in the manuscript. The authors have explored a novel mechanism of inhaled diesel exhaust particles (DEP) and HF diet can alter intestinal integrity and inflammation, which can be attenuated with probiotics. The only concern is the title of the manuscript looks not appropriate.  The title should be more attractive to readers, based on experimental results in the manuscript. 

Author Response

Reviewer 1 Comments:

The study entitled “Role of inhaled diesel exhaust particles and probiotics in intestinal integrity and inflammation in C57Bl/6 male mice” by Phillippi et al. is well designed and performed, and results are clearly presented in the manuscript. The authors have explored a novel mechanism of inhaled diesel exhaust particles (DEP) and HF diet can alter intestinal integrity and inflammation, which can be attenuated with probiotics. The only concern is the title of the manuscript looks not appropriate.  The title should be more attractive to readers, based on experimental results in the manuscript. 

Author response: thank you for your critique. Please see the updated title:

“Probiotics function as immunomodulators in the intestine in C57Bl/6 male mice exposed to inhaled diesel exhaust particles on a high-fat diet.”

Reviewer 2 Report

The paper by Philippi et al describes how exposure to inhaled diesel exhaust particles (DEP) and HF diet can alter  intestinal integrity and inflammation, and the possible attenuation by probiotics, in a murine model. The authors provided findings for different endpoints : the inhaled DEP stimulates globet cell formation in the small intestine, increases MUC-2 expression regardless of probiotic treatment,alters  tight junctions proteins  occludin ,claudin-3, and zo-1  and MMP-9. Probiotic treatment has also been shown to promote intestinal inflammatory response when exposed to inhaled diesel exhaust particles by analyzing TNF-alfa and IL-10 production.The authors propose that probiotics  can act as immunomodulators and serves to  normalize HF diet-mediated alterations in inflammatory responses

The study  aimed to determine if probiotic treatment could “protect” the gut microbiome from DEP exposure-mediated alterations in the intestines but is not conceived  a control group that is on a normal diet, so is not easy to discriminate the contribution of the high-fat diet to the effect of inhaled diesel exhaust particles

Author Response

The paper by Philippi et al describes how exposure to inhaled diesel exhaust particles (DEP) and HF diet can alter intestinal integrity and inflammation, and the possible attenuation by probiotics, in a murine model. The authors provided findings for different endpoints: the inhaled DEP stimulates goblet cell formation in the small intestine, increases MUC-2 expression regardless of probiotic treatment, alters tight junctions’ proteins occludin, claudin-3, and zo-1 and MMP-9. Probiotic treatment has also been shown to promote intestinal inflammatory response when exposed to inhaled diesel exhaust particles by analyzing TNF-alfa and IL-10 production. The authors propose that probiotics can act as immunomodulators and serves to normalize HF diet-mediated alterations in inflammatory responses

The study aimed to determine if probiotic treatment could “protect” the gut microbiome from DEP exposure-mediated alterations in the intestines but is not conceived a control group that is on a normal diet, so is not easy to discriminate the contribution of the high-fat diet to the effect of inhaled diesel exhaust particles.

Author response: Thank you for the critique. We recognize that this is a limitation to this study and have included this limitation in the Discussion section: “The authors also note that all of the study animals were on a HF diet, without a low-fat diet control for comparison, which limits the understanding of the outcomes in the intestines resulting from diet vs. exposure, and how these may be impacted through probiotic use.”  It is important to note that these endpoints were investigated following previous analyses in these same animals indicating that the high fat diet animals had exacerbated responses within the gut and lung microbiome, as well as upregulated indicators of cardiovascular disease. Therefore, we were particularly interested in what was happening at the intestinal barrier, as well as molecular signaling pathways, which may provide mechanistic insight into our previous findings of altered gut microbiota profiles in animals exposed to DEP on a high-fat diet. We plan to repeat this type of study in a low-fat model and do a comparative analysis in the future to further investigate the role of diet vs. exposure (as well as diet x exposure interactions). Regardless, we believe that these findings are important in understanding the complex relationship that the microbiome and gastrointestinal tract have in response to various insults, such as diet and inhaled toxins.

Reviewer 3 Report

In this study authors investigated the hypothesis that exposure to inhaled diesel exhaust particles (DEP) and high fat (HF) diet can alter intestinal integrity and inflammation, which can be mitigated with probiotics.

Their results confirmed that DEP exposure together with HF diet alters intestinal integrity and inflammation. They also proved probiotics fundamental role in understanding the influence of the microbiome in protecting and altering inflammatory responses in the intestines following exposure to inhaled DEP.

The manuscript is well-written and easy to understand. The topic is interesting and relevant in the field of air pollution exposure and gastrointestinal (GI) diseases however I have some observations:

Major:

  1. The number of mice used is unclear, in line 123 it is written that 48 mice were randomly assigned to be exposed via oropharyngeal aspiration (OA) to 35 μg diesel exhaust particles suspended in 35 μl 0.9% sterile saline (n=36), or sterile saline only (CON, n=24) so the total number should be 60 and not 48. Pleasy clarify.

  1. Line 124 the authors should explain the reason why they chose to use the concentration of 35 μg of diesel exhaust particles.

  1. Line 143 the authors should explain the reason why they chose to use 0.3g/day of probiotic.

  1. Line 214 why did the authors perform real-time RT-qPCR in only 8 samples?

  1. Line 223 Table 1: TNF-α and IL-10 primers are missing and please add GAPDH abbreviation in table 1 legend.

Minor:

  1. Line 14 please eliminate the word “male” that is repeated in line 15.

  1. Line 165 please substitute um with µm.

  1. Line 214 please substitute “RNA was extracted” with “RNA extraction”.

  1. Line 804 please eliminate one of the two “doi:”

  1. Authors should reduce self-citations.

Author Response

Reviewer 3 Comments:

In this study, authors investigated the hypothesis that exposure to inhaled diesel exhaust particles (DEP) and high fat (HF) diet can alter intestinal integrity and inflammation, which can be mitigated with probiotics.

Their results confirmed that DEP exposure together with HF diet alters intestinal integrity and inflammation. They also proved probiotics fundamental role in understanding the influence of the microbiome in protecting and altering inflammatory responses in the intestines following exposure to inhaled DEP.

The manuscript is well-written and easy to understand. The topic is interesting and relevant in the field of air pollution exposure and gastrointestinal (GI) diseases however I have some observations:

We thank the Reviewer for the comments provided.  Please see the responses following each critique below.

Major:

  1. The number of mice used is unclear, in line 123 it is written that 48 mice were randomly assigned to be exposed via oropharyngeal aspiration (OA) to 35 μg diesel exhaust particles suspended in 35 μl 0.9% sterile saline (n=36), or sterile saline only (CON, n=24) so the total number should be 60 and not 48. Please clarify.

Author response: This was a typo and has been updated in the manuscript. To clarify - n=24 for DEP and n=24 CON.

  1. Line 124 the authors should explain the reason why they chose to use the concentration of 35 μg of diesel exhaust particles.

Author response: Please see the following paragraph in the Discussion section: “This study aimed to determine if probiotic treatment could “protect” the gut microbiome from DEP exposure-mediated alterations in the intestines. Our results revealed that the microbiome plays a vital role in mediating these systemic inflammatory responses following a 30-day DEP exposure and HF diet protocol. The amount of clearance of the DEP from the lungs and subsequent ingestion via the mucociliary tract is still not fully understood, so we can only speculate on the direct and indirect influence of the DEP particles on gut bacteria. Previous ambient PM and vehicle exhaust exposure studies have reported alterations in the gut microbiome profile and systemic inflammation [33,34]; however, the effects of traffic-generated PM, such as DEP, on these outcomes is not as well characterized.  When adjusted for mice, the average DEP dosage of ~10ug/mouse daily is expected to be 40-fold higher than the comparable alveolar deposition from a 24-h inhalation of 100 ug/m3 in humans [82]. The dose of DEP used in the current study may be higher than that typically experienced in an ambient environmental scenario; however, we chose this dose based on inflammatory outcomes and alterations in microbiota profiles previously described in the lung and cardiovascular system of wildtype mice [36, 37, 82]. In doing so, it allowed for comparisons of outcomes described at the molecular and tissue levels in the gut to those described systemically and within the cardiopulmonary system.  We also recognize that the DEP utilized for this study is not necessarily representative of all of the diesel engine-generated PM. Nevertheless, this study provides preliminary information on the outcomes of subacute DEP exposure and diet-mediated alterations on gut microbial profiles and associated systemic inflammatory signaling pathways.”

  1. Line 143 the authors should explain the reason why they chose to use 0.3g/day of probiotic.

Author response: This has been updated in the methods section “Probiotic administration via drinking water was chosen over oral gavage to minimize stress to mice. Dosage calculation was determined by reference to a previous study using the same probiotic formulation delivered via drinking water in a rodent model, which resulted in significantly decreased inflammatory signaling in rodents fed a high-fat diet [38].”

  1. Line 214 why did the authors perform real-time RT-qPCR in only 8 samples?

Author response: There was a total of 12 animals/group in this study. To ensure we were consistent with the region of the intestines used for both protein and RNA quantification, we reserved 4 samples to be fixed for histological analysis, thus leaving 8 samples for RT-qPCR analysis.

  1. Line 223 Table 1: TNF-α and IL-10 primers are missing and please add GAPDH abbreviation in table 1 legend.

Author response: TNF-α and IL-10 mRNA expression was not quantified (only the expression at the protein level), therefore no primers were used. GAPDH abbreviation has been updated.

Minor:

  1. Line 14 please eliminate the word “male” that is repeated in line 15.

Author response: This has been edited for correction in the revised manuscript.

  1. Line 165 please substitute um with µm.

Author response: This has been edited for correction in the revised manuscript.

  1. Line 214 please substitute “RNA was extracted” with “RNA extraction”.

Author response: This has been edited for correction in the revised manuscript.

  1. Line 804 please eliminate one of the two “doi:”

Author response: This has been edited for correction in the revised manuscript.

  1. Authors should reduce self-citations.

Thank you, we have removed one of the self-citations in the methods section. This leaves 4 self-citations. The authors feel that the three citations, [34] Fitch et al., 2020, [36] Daniel et al., 2021, and [37] Phillippi et al., 2022 are important to include in this manuscript as these, collectively, report the effects of inhaled traffic-generated air pollutants on the gut and lung inflammatory profiles.  Furthermore, [36] Daniel et al., 2021 and [37] Phillippi et al., 2022 report findings in the same animals and exposure/treatment study as used in the current manuscript, which report exposure and/or diet mediated alterations in microbiota profiles in the lung and the gut. The final citation ([39.]. Suwannasual et al., 2019) is used to reference a previously established methods protocol.

Round 2

Reviewer 2 Report

The control group is needed both to state that inhaled particles have an effect on the intestine of the HF group, and also to affirm that probiotics can be used as immunomodulators in the hf diet group. Before including the low-fat diet group, it would be useful to have a normal diet control group.